# Stress Corrosion Cracking Behavior of Fine-Grained Al5083 Alloys Processed by Equal-Channel Angular Pressing (ECAP)

**DOI:** 10.3390/molecules26247608

**Published:** 2021-12-15

**Authors:** Asiful H. Seikh, Muneer Baig, Ateekh Ur Rehman, Faraz H. Hashmi, Jabair A. Mohammed

**Affiliations:** 1Centre of Excellence for Research in Engineering Materials, Deanship of Scientific Research, King Saud University, P.O. Box 800, Riyadh 11421, Saudi Arabia; mbaig@psu.edu.sa; 2Engineering Management Department, College of Engineering, Prince Sultan University, P.O. Box 66833, Riyadh 12435, Saudi Arabia; 3Department of Industrial Engineering, College of Engineering, King Saud University, P.O. Box 800, Riyadh 11421, Saudi Arabia; arehman@ksu.edu.sa; 4Department of Mechanical Engineering, College of Engineering, King Saud University, P.O. Box 800, Riyadh 11421, Saudi Arabia; 439106627@student.ksu.edu.sa

**Keywords:** ECAP, SCC, SSRT, tensile, SEM, EBSD

## Abstract

In the present study, the stress corrosion cracking (SCC) behavior of ECAP Al5083 alloy was investigated in air as well as in 3.5 % NaCl solution using the slow strain rate tensile test (SSRT). The characteristics of grain boundary precipitates (GBPs), specifically the microchemistry of the SCC behavior of Al5083 alloys, both in “as-received” condition and when deformed by the ECAP process, were examined. The correlations between the SCC resistance and GBP microchemistry were examined. A microstructural evaluation was performed using an optical microscope. SCC tests were carried out using a universal tensile testing machine and the fracture surfaces were studied using scanning electron microscopy (SEM). A strain rate of 1×10^−6^ s^−1^ was applied for the SSRT. As the passes increased, the SCC susceptibility of the fine-grained ECAP Al5083 alloy also increased. Moreover, higher ultimate tensile strength and greater elongation were observed. This was due to grain refinement, high-density separations, and the expanded extent of high-density dislocations instigated by severe plastic deformation. Due to the high strength and elongation, the failure analysis showed a ductile mode of fracture. Electron backscattering diffraction (EBSD) analysis was performed to determine more clearly the nature of cracking. EBSD analysis showed that the crack propagation occurred in both transgranular and intergranular modes.

## 1. Introduction

Aluminum alloys are used in a wide range of fields, such as in aerospace, marine transportation, shipping, cycling, and automobiles, due to their low density, high specific strength, excellent weldability, high corrosion resistance, and other functional characteristics. Aluminum (Al) alloys are widely used for castings in industry, since they have a low coefficient of thermal expansion, high hardness and wear resistance, high tensile strength at elevated temperatures, and good castability. Furthermore, 5083 aluminum alloy is one of a group of Al-Mg series aluminum alloys, in which Mg plays a major strengthening role through solid solution strengthening. This 5083 aluminum alloy is a passivating alloy, and its corrosion resistance in seawater largely depends on the passive film on its surface. Impurity anions in corrosive mediums can contribute to the corrosion of aluminum alloys (Soltis, 2015), and the strong erodibility of Cl^−^ and the presence of second-phase metal leads to pitting corrosion of aluminum alloys in seawater. Additionally, high-strength Al combinations do not offer good protection from stress corrosion cracking (SCC). This is especially true at the time of top strength for Al alloys [1,2,3], whereby precipitation solidification is responsible for the SCC weakness. The interaction of SCC in Al alloys is unpredictable, relying upon numerous boundaries. When a break occurs, its development is impacted by the microstructure of the material (heat treatment, pollution, foundation, and so forth), the degree of stress, and the state of the climate. The microstructural condition of the material is the most contemplated boundary used to discover microstructures impervious to SCC while introducing the elite mechanical properties of the material [1,4].

Aluminum alloys are among the most versatile and used materials for several technological purposes, mainly due to their good combination of intermediate-to-high mechanical strength with low density, good formability, and high corrosion resistance [5,6]. It is known that mechanical strength and corrosion properties can be modified through the use of an appropriate microstructure design. The use of intense grain refinement, down to the sub-micrometric scale (<1 μm) through severe plastic deformation (SPD), has been shown to be an effective strategy for improving the mechanical properties [7,8,9,10]. Equal-channel angular pressing (ECAP), one of the most well-known SPD techniques, is an efficient way to produce materials with ultrafine-grained (UFG) structures and intense shear plastic deformation [11]. Early investigations showed that ECAP is very effective process for refining grains in the microstructure and improving the mechanical properties of Al alloys [12,13,14,15]. Moreover, ECAP may introduce a number of microstructural heterogeneities, namely highly deformed areas; fragmentation and redistribution of intermetallic particles, crystallographic texture, and high internal stress [16,17].

It is generally recognized that petrol and flammable gas frameworks can be defiled with arrangements that are extremely forceful, such as in various preparations and aluminums, which are utilized in the vehicle and in the handling of oil-based commodities. It has been known for many years that up to one-quarter of hardware issues in the petrol refining industry are related with, stress consumption breakages, SCC, and hydrogen harm [18]. In general, austenitic stainless steel experience SCC failure, and many studies have reported on this material. However, other alloys or microstructures may also experience this failure. Despite the fact that SCC of austenitic steel is the most widely recognized issue, steels with other microstructures can also go through SCC. Likewise, SCC failure is additionally revealed in aluminum and its subordinate amalgams [19,20,21]. It is additionally generally recognized that aluminum amalgams that contain large amounts of dissolvable alloying components, particularly copper, magnesium, silicon, and zinc, are defenseless against SCC. They can fail by breaking along the grain limits under explicit conditions involving adequate amounts of stress. Notable conditions that can trigger SCC incorporation, however, are not restricted to water, but also include watery arrangements, acids (such as corrosive acids), natural fluids, fluid metals, and salt arrangements. Stresses adequate for breakage can be far below the burdens needed for net yields. Dissimilar to high-strength materials such as preparations, papers examining the SCC of Al are fairly scarce [20,21,22,23]. While more papers center around aluminum alloys in different destructive arrangements, not many of them center around the impacts of heat treatments or microstructures [24].

Researchers have shown that the primary cause of SCC in aluminum alloys is anodic dissolution, while for the 7xxx series of aluminum alloys the SCC is caused by hydrogen embrittlement [25,26]. Previous studies have mostly focused on the SCC behavior of different types of Al alloys, including chloride solutions [27,28,29,30,31,32], as well as in exfoliation corrosion environments [9,10,11]. Alexopoulos et al. [33] studied the effect of corrosion exposure on ultra-thin aluminum sheets and observed that for low-acceleration corrosion exposure, no surface deterioration existed. Hence, the corresponding degradation of the mechanical properties was assumed to be caused by hydrogen diffusion, whereby the hydrogen front advances with the corrosion front along the intergranular paths [34]. Alexopoulos et al. [35] reported that hydrogen embrittlement contributed to 27% of the ductility decrease, while micro-crack formations caused the remaining 73%. As reported, the observed increase in the Cu contents of the precipitates with aging correlates with the increasing resistance to intergranular SCC for Al 7075 alloys [36]. Xie et al. [37] obtained results in favor of the fact that stress corrosion cracking resistance is significantly improved by step quenching and ageing heat treatments. The influence of Al alloy precipitates [28], their compositions [29], the different testing parameters [27,30], and the roles of hydrogen and corrosion [38,39,40] were useful in revealing the mechanism of SCC for high-strength Al alloys. Fuente et al. [41] studied the peculiarities of the long-term corrosion of aluminum in industrial environments and concluded that the corrosion is influenced by the following factors: (a) the formation of highly cracked bayerite layers of a certain thickness; (b) the abundant formation of pitting on the base aluminum; (c) the abundant formation of amorphous aluminum sulfates throughout the thickness of the corrosion product layer; (d) the presence of chlorine in the bottom of the formed corrosion pits. Solution heat-treated samples exhibited higher resistance to oxidation than cast samples in all corrosive environments [42]. Stratmann and Streckel [43] showed that the potential drop across the oxide scale formed on top of the iron surface is small if the surface is kept in a humid environment (e.g., 95% relative humidity). Different solution types and concentrations, immersion times, and environments affect the different types of electrochemical reactions [44], corrosion product accumulation, localized corrosion (such as pitting corrosion initiation and further propagation [45]), and the production and adsorption of hydrogen [46], which absolutely impact SCC behavior. Cheng et al. [44] observed that the solute diffusion technique and oxygen reduction process control the layer thickness, which is associated with the electrochemical behavior of Al alloys below the thin layer of electrolytes. Zhang et al. [47] observed that for localized corrosion, such as pitting corrosion, the pit initiation was inhibited by the presence of air, but simultaneously the pit growth was accelerated as compared to the solution, which was in bulk amount. The comparable mechanism for Al aluminum in a solution of a thin electrolytic layer of Na_2_SO_4_ was shown by Zhou et al. [48]. Many methods have been used to improve the corrosion resistance of Al alloys, including the addition of trace elements to control the recrystallization during heat treatment [49,50,51,52]. On the contrary, other researchers [51] observed that in salt spray environments, corrosion may appear as surface build-up. Liao et al. [53] reported that the copper corrosion rate increases as the thin electrolyte layer (TEL) thickness decreases during the initial stages. Researchers studying the influence of hydrogen on the propagation of intergranular corrosion (IGC) defects of 2024 aluminum alloy concluded that H did not influence the elementary processes of IGC, although it had a significant impact on the local reactivity [54]. In addition to these factors, the evolved hydrogen seems to play a very important role in the degradation of the materials [34,35,55]. Larignon [55,56] found that for the cyclic polarization test, the influence of hydrogen corrosion on the Al alloys was more pronounced. This may be attributed to the higher exchange rate on the surface area during the cyclic polarization tests. The literature also reports that deformation of materials also tends to affect the SCC behavior in sulfide environments. The importance of electrochemical-induced hydrogen, anodic dissolution, and their combined effects in causing SCC in high-yield Al alloys has been reported in the literature. Evolved hydrogen not only corrodes the material, it is also responsible for inducing stress. In the present study, a quantitative assessment of the degradation of Al alloys by hydrogen has been made rather than taking the simpler qualitative approach. Researchers have relentlessly worked on establishing the relationship between the grain boundary and grain size with the electrochemical corrosion behavior of materials.

In this present study, Al5083 alloy samples underwent ECAP processing with 1, or 3 passes. Here, the grain refinement of Al5083 allows occurred after going through different ECAP passes, after which the SCC resistance was investigated by SSRT method in both air and 3.5% NaCl environments. This SSRT method is considered as an advanced testing method. Using this process, the SCC susceptibility of most of the alloys and materials could be measured [57]. The present research aimed at investigating how the SCC behavior of AL5083 alloy is affected by ECAP passes. Additionally, an attempt was made to correlate SCC with the microchemistry of GBPs.

## 2. Materials and Methods

Commercially available Al5083 alloy was used for studies. The chemical composition of this alloy is given in Table 1.

To test the mechanical properties and SCC characteristics, tensile samples were cut perpendicular to the rolling direction from Al alloy sheet plate, as shown in Figure 1.

The resulting sample exhibited a tensile axis perpendicular to the direction in which the SCC susceptibility of the sample was maximal. To start the tests, the surfaces of the specimens were sequentially polished with emery paper (SiC) up to a fineness of P2500. These specimens were then cleaned using alcohol to remove any moisture that might have settled on them. Silica gel was used to cover the specimen surface, except for the middle portion, with a gauge length of 20 mm (Figure 1B). Subsequently, tensile specimens were exposed to 3.5% NaCl solution atmosphere, as this is the most common solution used to conduct these kinds of tests (Figure 1C). The tension test specimens were pulled in the short-transverse orientation so that the load axis was perpendicular to the rolling direction of the original sheet materials, the same as the fracturing orientation of specimens. The tension test specimens representing each sensitization condition from the complete set of sensitized AA5083 alloys tested in lab air at an extension rate of 0.25 in./in./min (0.25 cm/cm/min). The results in terms of the yield strength, ultimate tensile strength, and strain to failure were recorded. This was followed by a slow strain rate test (SSRT), which was conducted while maintaining a constant strain rate of 10^−6^/sec. To conduct these tests, a WDML-40kN Material Test System was employed. ASTM standard G129 was followed while conducting these tests [57]. The corrosive atmosphere was maintained such that it would replicate the corrosiveness of 3.5 wt.% NaCl, pH 7.0, kept at 25 ± 2 °C. No imposed polarization was applied during the SSRT tests; these tests were carried out at open-circuit potential. Thus, the total corrosion time before the fracture of the SSRT Al alloy was calculated. All the while, the same strain rate (10^−6^/s) was maintained to assess degradation in the mechanical properties. On completion of the test, the specimen was carefully detached without damaging the corrosion-affected surface. To check the reproducibility, each experiment was repeated 4 times. The energy density, elongation, and ultimate tensile strength (UTS) were also obtained from the stress–strain curves from the test data. Researchers [42,43] have found that the rate of electrochemical corrosion decreases with the decrease in grain size for high-purity aluminum. The formation of microcells between the grain boundary precipitates and the PFZ tends to make the grain boundaries more susceptible to corrosion [44]. Dan et al. [45], during their study on grain boundary corrosion, observed a considerable difference in the corrosion resistance of fibrous grains with respect to recrystallized grains in strained microstructures. They concluded that low-angle grain boundaries were less impacted by intergranular corrosion. Liu et al. [46] observed that the high-angle grain boundaries (HAGBs) were the common sites prone to intergranular corrosion, as a result of which cracks propagated via these regions. This phenomenon of crack movement along the HAGBs is a common occurrence in recrystallized grains [47]. Hence, fibrous grains exhibited better corrosion resistance than the more susceptible recrystallized grains [48].

After ECAP processing was performed, the samples were ultrasonically cleaned. The ECAP samples were properly polished using different grades of emery paper followed by cloth polishing in which alumina were used as abrasives. Microstructural analysis of these samples was performed using a Leica optical microscope (Leica Microsystems, Wetzlar, Germany).

A hot chronic acid bath was used to clean the fractured SSRT test. This bath was composed of 180 g/L CrO_3_. Then, the fractured surface characteristics were examined using SEM. All studies were carried out using a JEOL JSM-6360 instrument (JEOL Ltd., Tokiyo, Japan).

The specimen was machined and further polished mechanically to produce scratch-free and smooth surfaces before texture measurements. The texture test was conducted on the rolling samples’ 10 × 10 mm^2^ surfaces. In an FEI Quanta FEG 250 FESEM, the microstructures of the specimen were analyzed using electron backscattering diffraction (EBSD).

## 3. Results and Discussions

### 3.1. Microstructure Analysis

The microstructures of as-received and ECAP samples are given in Figure 2. The original alloy has a coarse grain structure. The microstructures demonstrate the fact that with a high deformation degree, microscopically homogeneous fine-grained structures are formed. A grain size of 145 µm can be observed for the as-received sample. From there, the grain size decreases to 37 µm after the 3rd pass. After the 1st and 2nd passes, the grain size reduces to 98 µm and 56 µm, respectively. The grain size determination and analysis were performed using image J software. In a previous study [58], it was shown that there was a huge expansion in elasticity of the material after ECAP. The distortion created by ECAP results in grain refinement.

In Figure 2, it can be seen that elongation occurs, while refinement of the grain is also visible. After an initial pass, there is a reduction of the grain size as the unique grains in the specimen separate into subgrain groups.

By virtue of multi-pass ECAP, for the 1st pass, the speed of dislocation expansion is significantly greater as compared to the dislocation decimation in light of the meagre dislocation density. Hence, the grains of the material are mostly refined. During subsequent passes pressing interactions, the grain refinement is proceeded considering the way that the disengagement thickness and the inner energy are at this point extended. Nevertheless, the development of the inward energy causes translucent recuperation and recrystallization shapes, meaning the grain refinement is decreased bit-by-bit after a couple of passes. On the other hand, the grain refinement advancement system in the example changes for the different squeezing courses. The gathering and evening out of the disengagements are similar to course C. Through the examination of the disengagement progression, it can be seen that nanostructured materials can be obtained using the ECAP process. The process of refinement of the grain may be depicted as a steady, remarkable recovery and recrystallization process. The grain refinement procedure controls the dynamic equalization due to age and destruction of the dislocations [59].

### 3.2. Slow Strain Rate Tensile Testing

Figure 1 shows a schematic diagram of the stress corrosion cracking mechanism. The effects of environmental conditions along with the underlying mechanism of hydrogen embrittlement and corrosion are studied in the present research. Compared to the corrosive environment, the total elongation of Al5083 is larger in air. Additionally, the specimen shows greater elongation in the air environment in comparison to the elongation in 3.5% NaCl solution. Once the elastic limit of the material has been reached, plastic deformation occurs, followed by a hardening, whereby the stress must be continuously increased in order to continue producing plastic deformation. Localized corrosion pits and surface cracks are the primary causes of cracking under strain. The surface crack dimensions (length and depth) in the air specimen are greater in contrast to the specimen in the solution environment.

According to Alexopoulos et al. [33,35,38,39], the cross-sectional area of the specimen representing the unaffected zone of the Al alloy was denoted as the ‘effective thickness’. Kamoutsi et al. [34] reported that with the decrease in ‘effective thickness’, a small reduction in the UTS is observed in corroded tensile Al alloy specimens. This parameter reverts back to its original state when the corroded layer is removed [34]. In this research work, the primary corrosion product was observed. These seemed to be distributed independently in the corroded layer, causing the affected layer to remain mostly intact. The defects resulting from corrosion affected the UTS and YS (yield strength), although both UTS and YS were the same as for the Al alloy specimen in laboratory conditions. It should be noted that the reduction in ‘effective thickness’ does not indicate entire elongation, as the core difference is in the mechanical properties. Even after the complete removal of the corrosion layer, the elongation of the control samples was smaller than that of the corroded samples. Additionally, the removal of the corrosion was not able to restore the total deformation (elongation). The trapping of the hydrogen and the dissimilarity in the mechanical characteristics were due to the deep penetration of hydrogen in the corroded regions. Comparing the as-received sample with the more deformed sample, there were more and deeper cracks, which provided a higher volume of trapped hydrogen on account of the higher contacting surface area and the blocked environment [60]. From the results, we could predict and describe the main role of the electrochemical reaction induced by hydrogen in the environment. Initially, micro-cracks were initiated in the subsurface area of the corrosion network under stress, which was the reason for the stress concentration, then the subsurface attacks grew as the depth increased (Figure 3A). For the as-received sampled, as compared to the deformed sample, the attacks of the subsurface areas were greater, and these corrosion attacks developed to a deeper depth and enhanced the sensitivity of SCC (Figure 3B,C). Corrosion developed beside the grain boundary linkage with the production of hydrogen in the electrochemical reaction (Figure 3A–C) [34,61,62].

The SSRT stress strain curve for the Al5083 alloy is introduced in Figure 4 and Figure 5, while Table 2 shows the yield strength, UTS, elongation, and SCC index (I_scc_) values for the four distinct examples. Obviously, in contrast with the as-received sample, the yield strength and UTS were upgraded after ECAP processing to a great extent, starting from the reinforcing impact of the refined grains [52]. Furthermore, samples after the third pass exhibited good properties, with a mix of strength and elongation, both in air and 3.5% NaCl, in contrast with the other two examples processed by ECAP. The moderately favorable third pass ECAP samples were essentially ascribed to grain refinement and the similarly homogeneous structure. A remarkable resemblance among every one of the specimens was seen for the critical loss of UTS and elongation under 3.5% NaCl conditions in contrast with air conditions.

The SCC susceptibility index (I_scc_) represents the stress corrosion susceptibility of the Al alloy, which is calculated by Equation (1) [63]:I_scc_ = (1 − *ɛ_solution_*/*ɛ_air_*) × 100%(1)
where *ε_solution_* and *ε_air_* are ε_f_ in 3.5 wt.% NaCl solution and air, respectively. It is considered that the material under testing conditions is resistant enough to SCC if the value of I_scc_ is near 1. Here, all I_scc_ values are presented the in form of percentages in Table 2.

The I_scc_ values of SCCs are determined using Equation (1). It is very obvious that the as-received Al5083 alloy displays high SCC susceptibility, with a SCC value of 26.12% in 3.5% NaCl. After ECAP processing, all the ECAP specimens show higher Iscc, demonstrating a more noteworthy inclination of SCC. After the 1st pass, the I_scc_ decreases slightly, although with subsequent passes these values increase gradually. This proves that the SCC susceptibility properties of the Al alloy increase with ECAP processing.

SEM micrographs (Figures 7 and 8) revealed that the crack growth was divided into two stages, namely the steady-state crack growth stage and the incubation crack propagation stage. The steady-state stage corresponds to the hydrogen embrittlement mechanism, while a dissolution-controlled mechanism is found to be associated with the crack propagation stage. The dissolution-controlled phase, which is the initial phase, is presumed to cause the same amount of corrosion as in the 3.5 % NaCl solution conditions. However, there is an enormous difference in total elongation with the SSRT treatment of the as-received and deformed specimens. Regarding the tensile stress, different types of microcracks are formed in the bottommost areas of corrosion defects, causing breakage of the passive oxide film layer and penetration of the electrolytes into crevices. The crack tip of the testing specimen is dissolved, and with the electrochemical hydrolysis reaction an acidic environment is generated. Then, the Al alloy is corroded and dissolved as anodic sites and hydrogen bubbles are observed in the corrosion pit [64,65]. The crack tips show regions with precipitated Al(OH)_3_ with a steady pH [65], while the hydrogen generated in NaCl solution is related to the SCC [66]. The evolution of hydrogen ions (H^+^) results in the penetration of a certain percentage of the hydrogen ions into the Al specimen, causing embrittlement in the Al alloy [34,66,67]. Najjar et al. [68] showed that defects were generated in the grains due to anodic dissolution, which promoted the discharge of hydrogen ions and their entry into the metal, resulting in embrittlement of the specimen in NaCl solution. Hence, it may be concluded that the combined effects of hydrogen embrittlement, stress, and anodic dissolution (Figure 3D) result in SCC of Al alloy in the presence of chloride ions. This may be enhanced by stress, as validated by the smooth cleavage surfaces and transgranular fractures. In the absence of dynamic stresses, the effect of hydrogen is minimized. The as-received sample shows a different SCC mechanism than the deformed sample in the chloride solution. Similar to the corrosion test, the as-received tensile specimen shows a larger degree of corrosion than the ECAP specimen in the chloride environment. Large corrosion sites are formed when many individual pit groups merge by way of anodic dissolution. Hence, it may be fair to assume that the mechanical properties of the as-received and deformed samples differ due to the presence of obstructive environments at the corrosion sites. The SSRT specimen tested in normal ambient environment shows a slightly larger total elongation than when tested in the 3.5% NaCl solution. This shows that hydrogen permeation and facilitation of stress to corrosion are the primary reasons for stress corrosion cracking. Additionally, it can be observed that the hydrogen induced by corrosion is the dominant electrochemical phenomenon for the Al alloy. However, the ECAP samples show a lower loss of hydrogen-induced ductility as compared to the as-received sample. The main reason for the rounded pit in the 3rd pass ECAP sample is the corrosion front formed on the former pass, which is unfavorable as it causes an obstructed environment and prevents the adsorption of Cl^−^ ions.

### 3.3. Correlation between SCC Resistance and GBPs

ECAP significantly improved the stress corrosion cracking resistance of the Al 5083 alloy. It has been very well proven that the composition of GBPs and their morphology, primarily their precipitation content, are the predominant causes of SCC resistance. For the as-received and deformed samples, the potential difference in GBPs was found to be higher than the matrix difference. GBPs form more galvanic microcells, leading to a higher amount of microgalvanic corrosion [53,54,55]. As can be seen in Figure 6, the initial microstructure showed fine precipitates homogeneously distributed in the sample. The microstructure analysis showed successful grain refinement and the presence of subgrains inside the grain, as expected after the initial ECAP passes. Figure 6 shows the TEM micrographs of as-received and ECAP sample. The figure shows the presence of subgrains that were formed as a consequence of the deformation process. The presence of dislocation tangles after two extrusion passes and the characteristic shape of the Mg_2_Si precipitates were also observed in this figure. The phase identification has been done by XRD technique and the XRD graphs are given in the Appendix A. 

In addition, an uninterrupted stretch of the grain boundaries that are under constant stress form a continuous anode electrochemical reaction network, which leads to the propagation of SCCs along these boundaries. The SCC resistance under constant stress was highly improved for ECAP samples as compared to the as-received samples. Consequently, during the SCC process, high-affinity microgalvanic corrosion was observed. As suggested, the modification of distribution of GBPs cannot significantly enhance the SCC resistance of the Al alloy.

### 3.4. Microstructural Analysis of Failed Surfaces

After tensile testing was performed, the fractured surfaces of the specimens were observed using SEM. The fractured surfaces are shown in Figure 7 and Figure 8. From the SEM images, it is quite clear that grain refinement occurs as the passes increase.

The fracture properties on the surfaces also change as the passes increase. The fracture is quite brittle by nature in the 1st pass sample compared to other samples. The fractures are more ductile in the 2nd pass and 3rd pass samples. It is also noticeable that the fractures become narrower and less shallow as the passes increase. This is because the strength of the alloy increases as the passes increase.

### 3.5. EBSD (Electron Backscattering Diffraction) Analysis

EBSD analysis was performed along the cross-sectional area of these four specimens to more clearly determine the nature of the cracking. The EBSD micrographs in air and in 3.5% NaCl solution are shown in Figure 9 and Figure 10, respectively. In the previous micrograph, given in Figure 8, it is quite clear that the cracking nature is more ductile as the passes progress, although with EBSD this becomes clearer. The distinction as to whether grain or subgrain sizes are measured is completely independent from the EBSD data file, depending only on user-defined misorientation threshold values. A value of 15° misorientation is commonly used to define HAGBs (high-angle grain boundaries), which separate individual grains in aluminum alloys [69]. In this work, LAGBs between 3° and 15° were identified as grain boundaries separating individual subgrains. The lower limit of 3° was chosen to ensure a certain distance to the measurement inaccuracy for the EBSD technique, which lies in the range of 1~2° and depends on the alloy composition and position of the focused electron beam inside the measurement area.

From these four structures, it is evident that the cracking mode and nature are both transgranular and intergranular in nature, although during the 2nd pass the cracking mode exhibits a slight transgranular nature. For EBSD measurements, the samples were slightly tilted at a certain angle, meaning the cracked morphologies shown in EBSD images are slightly different from the real morphologies, as shown in Figure 6. Moreover, the width of cracks is much larger than that of the grain boundaries. Therefore, it would be very difficult to confirm intergranular cracking on the surface of as-forged samples with a fine-grained structure. As the passes progress, grain refinement occurs, which has directly decides whether the crack propagation is trangranular, intergranular, or a mixture of both. Due to the smaller grain size in the high pass samples, cracks can easily meet with the grain boundaries. Thus, partially intergranular cracking can possibly occur in particular segments of some grain boundaries.

Regarding the EBSD results of the pole figures and misorientation maps of the surface layer, cross-sections and longitudinal sections of the ECAP samples are shown in Figure 10. The high-angle grain boundaries (HAGBs), namely misorientation angles above 15 degrees, and low-angle grain boundaries (LAGBs) are shown as thin red lines and thick blue lines, respectively. On the uppermost surface film (ED–TD) of the specimen, the grain size is very much heterogeneous, while most of the grain boundaries are scattered along the y-axis direction. Compared with the y-axis, the aspect ratio of grains within the longitudinal section (8.256) is considerably larger than that within the cross-section (3.895). Comparing the results, it can be observed that the GBs of the as-received samples are less dense than for ECAP samples. 

### 3.6. EBSD Results of Areas near the Stress Corrosion Cracks

Figure 10 shows EBSD maps of the stress corrosion cracking observed on the side face of the fracture of the longitudinal specimen. The cracks are marked by the thick black lines in Figure 10B–D. It can be observed in Figure 10A that the cracks are discontinuous and small, indicating typical intergranular cracking types. In Figure 10B, misorientation angles between the grains on both sides of the cracks were calculated using software to perform the EBSD analysis. The result shows that maximum sections of cracks propagated along with the HAGBs. As shown in Figure 10, the misorientation angles of more than 95% of the grains on the both sides of crack are larger than 10° and dominated by angles ranging between 50° and 60° (accounting for 36% of angles). Hence, the cracks propagate along the GBs, with a misorientation range of approximately 50°–60°.

## 4. Conclusions

The following conclusions were from the results: (1)ECAP increased the strength and elongation of the samples, which was clear from the microstructures, while further SCC studies also supported these results;(2)In both the solution and air environment, corrosion-induced surface cavities followed by the formation of the subsurface micro-crack network were observed on the Al5083 alloy samples;(3)ECAP enhanced the mechanical properties of the Al5083 alloy as compared with the as-received sample. However, an adverse effect was observed on the corrosion resistance of the ECAP sample as compared with the as-received sample;(4)Grain refinement was observed from the 1st pass to the 3rd pass. The as-received sample showed a grain size of 145 µm. From there, the grain size decreased to 37 µm after the 3rd pass;(5)The SCC susceptibility of Al5083 alloy was higher in 3.5% NaCl than that in air environment;(6)Grain refinement may provide more sites to initiate corrosion, thereby enhancing the corrosion rate and consequently decreasing corrosion resistance;(7)The stress corrosion cracking started from the side face and propagated to the center, while the corroded area featured intergranular cracking;(8)The stress corrosion cracks were small, discontinuous, and tended to propagate along the GBs, with misorientation of 50–60° between adjacent grains. The cracks of longitudinal specimens propagated relatively parallel to the tensile direction, while the crack propagation of transverse specimens zig-zagged and tended towards the direction of maximum shear stress.

## Figures and Tables

**Figure 1 molecules-26-07608-f001:**
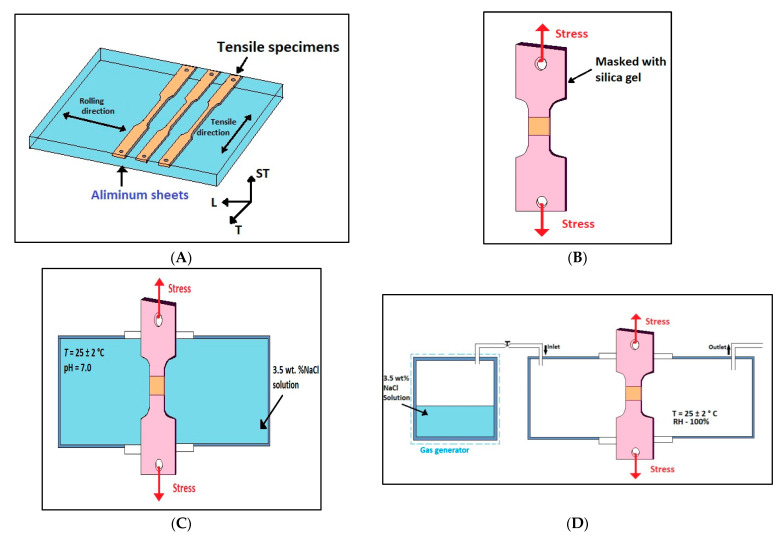
Schematic diagram of the sample processing (**A**) and preparation (**B**) for the SSRT tests, as well as the devices used in the tensile test for immersion in (**C**) air and (**D**) 3.5 wt% NaCl.

**Figure 2 molecules-26-07608-f002:**
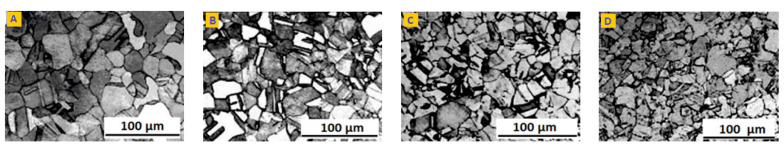
Microstructure (**A**) as-received and ECAP samples of Al5083 alloy after the (**B**) 1st pass, (**C**) 2nd pass, and (**D**) 3rd pass.

**Figure 3 molecules-26-07608-f003:**
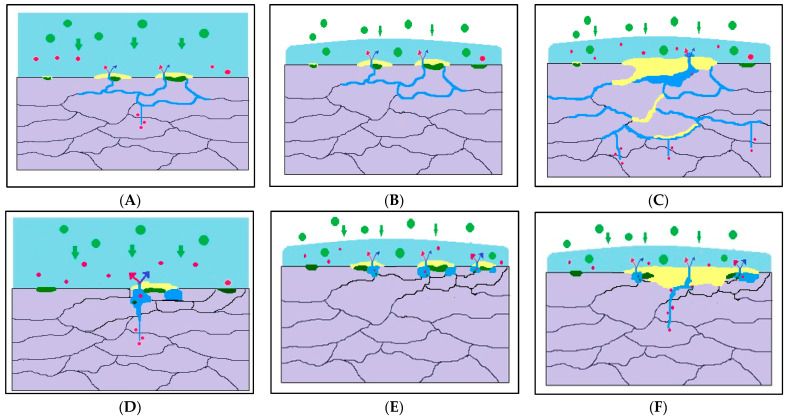
Schematic diagram of the corrosion and SCC processes of Al5083 alloy samples in 3.5 wt% NaCl solution and air environments: (**A**) microcracks are initiated under stress (**B**,**C**) subsurface attacks develop to a deeper depth (**D**) synergistic effect of stress, anodic dissolution, and hydrogen embrittlement (**E**) accelerates the anodic dissolution of aluminum alloy (**F**) Pitting groups coalesce together to form a larger corrosion site.

**Figure 4 molecules-26-07608-f004:**
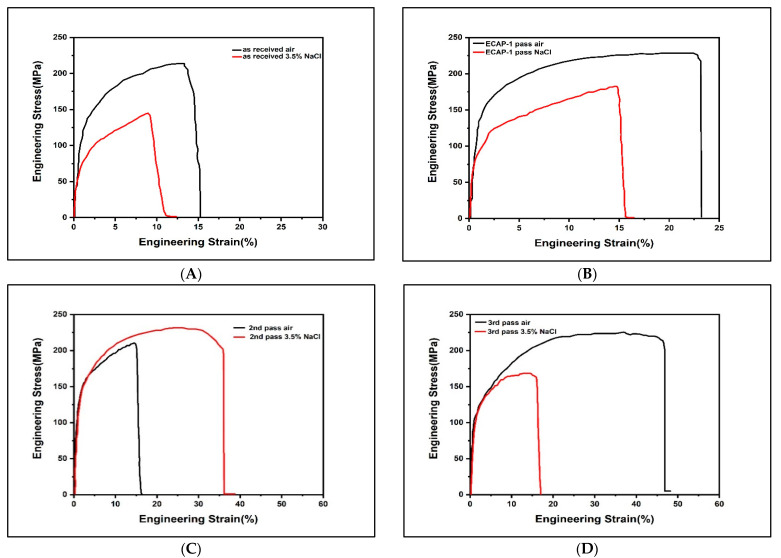
Stress–strain diagram of as-received Al5083 alloy (**A**) and ECAP after the 1st pass (**B**), 2nd pass (**C**), and 3rd pass (**D**) in air and 3.5% NaCl solution.

**Figure 5 molecules-26-07608-f005:**
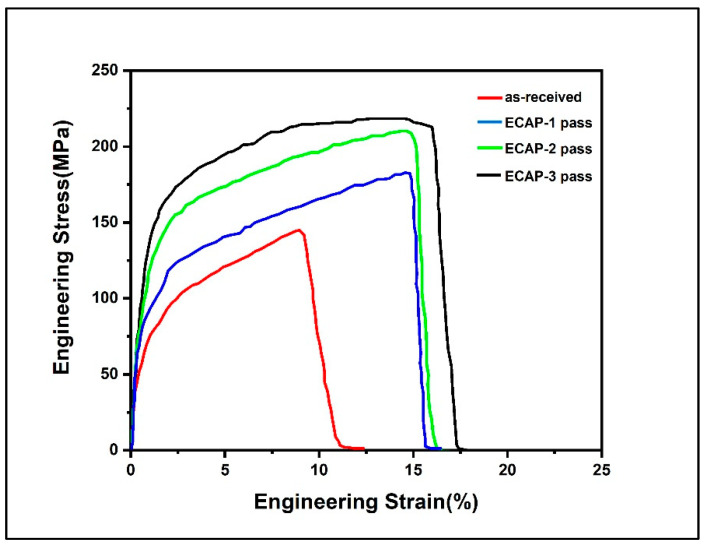
Comparison of stress–strain values of the samples tested in 3.5% NaCl solution with different passes.

**Figure 6 molecules-26-07608-f006:**
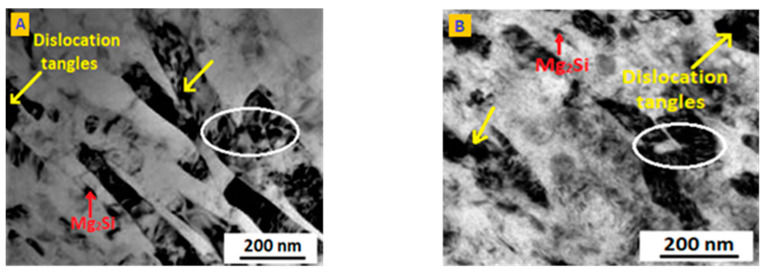
TEM images of Al5083 alloy in (**A**) as-received condition and after the (**B**) 1st pass, (**C**) 2nd pass, and (**D**) 3rd pass.

**Figure 7 molecules-26-07608-f007:**
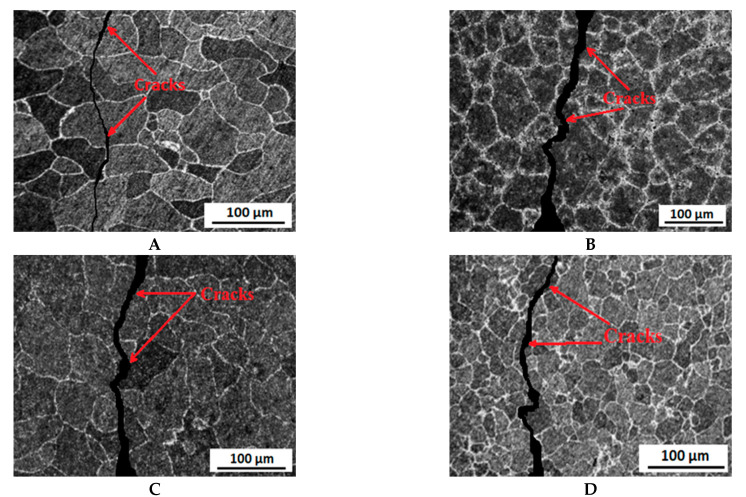
Microstructural analysis of the failed surfaces of Al5083 alloy tested in air for the (**A**) as-received sample and samples after the (**B**) 1st pass, (**C**) 2nd pass, and (**D**) 3rd pass.

**Figure 8 molecules-26-07608-f008:**
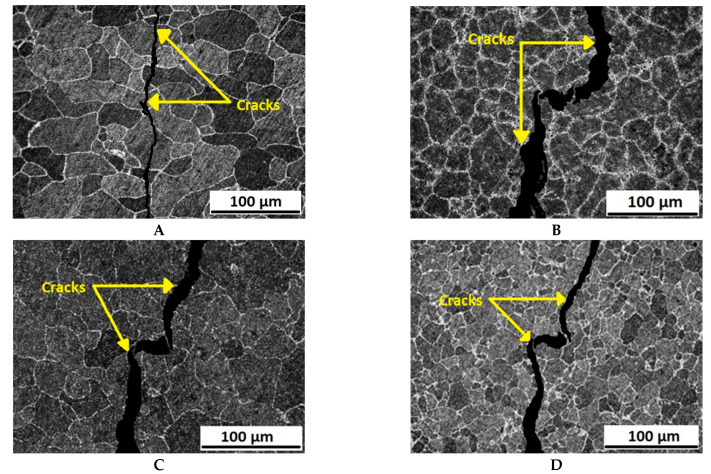
Microstructural analysis of the failed surfaces of Al5083 alloy tested in 3.5 wt.% NaCl solution for the (**A**) as-received sample and for samples after the (**B**) 1st pass, (**C**) 2nd pass, and (**D**) 3rd pass.

**Figure 9 molecules-26-07608-f009:**
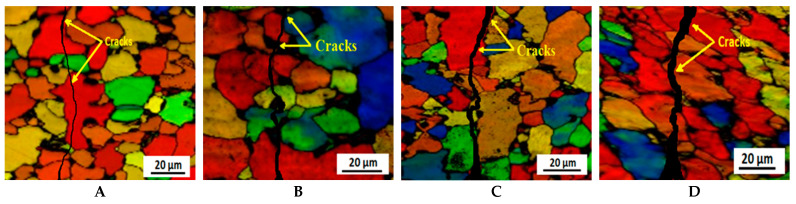
EBSD orientation micrograph of Al5083 alloy in air for (**A**) the as-received sample and samples after the (**B**) 1st pass, (**C**) 2nd pass, and (**D**) 3rd pass.

**Figure 10 molecules-26-07608-f010:**
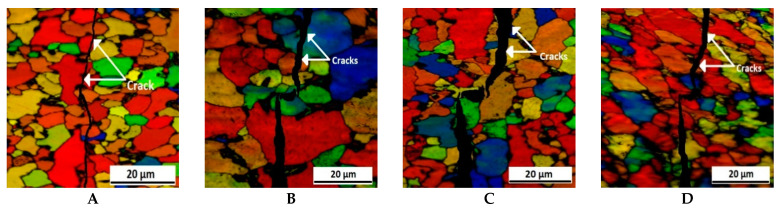
EBSD orientation micrograph of Al5083 alloy in 3.5% NaCl solution for (**A**) the as-received sample and samples after the (**B**) 1st pass, (**C**) 2nd pass, and (**D**) 3rd pass.

**Table 1 molecules-26-07608-t001:** Chemical composition of Al5083 alloy.

Element	% Amount
Si	0.4
Fe	0.4
Cu	0.1
Mn	0.7
Mg	4.3
Zn	0.25
Ti	0.15
Cr	0.20
Al	Balance

**Table 2 molecules-26-07608-t002:** Yield strength, UTS, elongation, and stress corrosion cracking (SCC) values of as-received and ECAP Al5083 alloy samples.

Samples	Condition	Yield Strength (MPa)	% Error	Ultimate Tensile Strength (MPa)	% Error	Elongation (%)	% Error	SCC Susceptibility Index, I_scc_ (%)
As-received	Air 3.5% NaCl	68 71	0.613 0.559	186 151	0.413 0.415	12.02 8.88	0.113 0.229	26.12
1st pass	Air 3.5% NaCl	153 138	0.526 0.529	267 212	0.626 0.346	11.86 8.96	1.026 0.665	24.50
2nd pass	Air 3.5% NaCl	106 148	1.002 0.884	235 235	0.202 0.545	12.50 9.01	1.029 0.721	27.92
3rd pass	Air 3.5% NaCl	96 155	0.479 0.564	261 246	0.779 0.665	15.41 10.77	0.559 0.452	30.11

## Data Availability

The experimental datasets obtained from this research work and the results analyzed during the current study are available from the corresponding author on reasonable request.

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
