# Peer review of "Stress Corrosion Cracking Behavior of Fine-Grained Al5083 Alloys Processed by Equal-Channel Angular Pressing (ECAP)"

_molecules, 2021, doi:10.3390/molecules26247608_

Round 1

Reviewer 1 Report

This manuscript aims to investigate the SCC resistance property of Al5083 alloy sample that has gone through several ECAP passes for refinement of grain structure.

The results could be of interest to the readers of this journal. The work is well presented and supported by sufficient scientific evidence. However, there are several technical and presentation issues needed to be further clarified/improved, including severe improvement in English language.

1. English needs to be improved. There are several grammatical errors throughout the entire manuscript. Some of them are appended in here.

2. The Introduction section lacks a statement on the novelty of this work.

3. Introduction part seems too pompous and elaborative as compared to the results/discussion part. It is advised to provide a brief introduction with fewer references.

4. Abstract section: Line 16:The sentence ‘The characteristics of grain boundary precipitates (GBPs), specifically microchemistry, on 16 the SCC of as received and deformed by ECAP process of Al5083 alloys were examined’ needs to be rewritten. English language error.

5. Abstract section: Line 23: The sentence ‘The outcomes were credited to grain refinement, high-density separations..’ needs to be checked. English language error.

6. Introduction part: Line 32: Rewrite the sentence ‘Aluminium (Al) alloy which is by nature high strength, are typically picked on ac-32 count for its good strength as well as firmness, that are gotten from precipitation solidify’-.

7. Line 351: ‘After tensile testing was done, the failed surfaces of the specimens were observed
351 using SEM.’. Replace the word “failed” with ‘fractured’.
8. Fig. 5(b) TEM micrograph lacks clarity. Replace it if possible.

9. How can the author confirm that the as-indicated particles (Fig. 5(a-d) TEM micrographs) are of Mg2Si precipitates? You need to perform XRD, HRTEM, SEM-EDX etc for exact phase identification.

10. The authors claim that Figure 9 (a) to (d) represent the inverse pole figure maps. However, Fig 8 also shows the same nature of EBSD micrographs. In verse pole figure maps need to describe/indicate Rolling and normal direction clearly. In verse pole figure maps and phase distribution maps in EBSD are not same. Authors need to work on this and provide meaningful micrographs.

11. Is Grain refinement the only reason for decreased corrosion resistance?

12. There is serious error in Figure numbers. Fig. 4 is repeated twice.

13. Figure 4 (or 5) Schematic diagram of the SCC process: Proper labelling is required for each diagrams.

14. Sentence 289: ‘SEM micrographs (Figure 5) revealed...’ Where is SEM micrographs in this work ?

15. Why two (same) numbers are used for Reference marking?

Author Response

Response to Reviewer 1 Comments

Comments and Suggestions for Authors

This manuscript aims to investigate the SCC resistance property of Al5083 alloy sample that has gone through several ECAP passes for refinement of grain structure.

The results could be of interest to the readers of this journal. The work is well presented and supported by sufficient scientific evidence. However, there are several technical and presentation issues needed to be further clarified/improved, including severe improvement in English language.

  1. English needs to be improved. There are several grammatical errors throughout the entire manuscript. Some of them are appended in here.

Ans. Necessary changes are done.

  1. The Introduction section lacks a statement on the novelty of this work.

Ans. Necessary changes are done.

  1. Introduction part seems too pompous and elaborative as compared to the results/discussion part. It is advised to provide a brief introduction with fewer references.

Ans. Necessary changes are done.

  1. Abstract section: Line 16:The sentence ‘The characteristics of grain boundary precipitates (GBPs), specifically microchemistry, on 16 the SCC of as received and deformed by ECAP process of Al5083 alloys were examined’ needs to be rewritten. English language error.

Ans. Changes are done.

  1. Abstract section: Line 23: The sentence – ‘The outcomes were credited to grain refinement, high-density separations..’ needs to be checked. English language error.

Ans. Changes are done.

  1. Introduction part: Line 32: Rewrite the sentence –‘Aluminium (Al) alloy which is by nature high strength, are typically picked on ac-32 count for its good strength as well as firmness, that are gotten from precipitation solidify’-.

Ans. Rewrited.

  1. Line 351: ‘After tensile testing was done, the failed surfaces of the specimens were observed 351 using SEM.’. Replace the word “failed” with ‘fractured’.

Ans. Replaced

  1. Fig. 5(b) TEM micrograph lacks clarity. Replace it if possible.

Ans. Replaced.

  1. How can the author confirm that the as-indicated particles (Fig. 5(a-d) TEM micrographs) are of Mg2Si precipitates? You need to perform XRD, HRTEM, SEM-EDX etc for exact phase identification.

Ans. XRD had been used. The XRD graph is given below.

                         As received                                                     ECAP 1

                       ECAP 2                                                          ECAP 3

  1. The authors claim that Figure 9 (a) to (d) represent the inverse pole figure maps. However, Fig 8 also shows the same nature of EBSD micrographs. In verse pole figure maps need to describe/indicate Rolling and normal direction clearly. In verse pole figure maps and phase distribution maps in EBSD are not same. Authors need to work on this and provide meaningful micrographs.

Ans. This is done by mistake. Necessary changes are done.

  1. Is Grain refinement the only reason for decreased corrosion resistance?

Ans. The detailed discussion is given in the paper.

  1. There is serious error in Figure numbers. Fig. 4 is repeated twice.

Ans. Necessary changes are done.

  1. Figure 4 (or 5) Schematic diagram of the SCC process: Proper labelling is required for each diagrams.

Ans. Descrition given in title section.

  1. Sentence 289: ‘SEM micrographs (Figure 5) revealed...’ Where is SEM micrographs in this work ?

Ans. The figure no. is already mentioned.

  1. Why two (same) numbers are used for Reference marking?

Ans. This is done by mistake…..necessary changes are done.

Reviewer 2 Report

This paper investigated the SCC behavior of 5083 Al alloy processed by Equal Channel Angular Pressing (ECAP). Although some results were reported and discussed, I cannot recommend the acceptance of this paper due to the following reasons.

(1) 5083 Al is a common Al alloy, and ECAP is not a novel processing method. I believe that plenty of studies about the ECAP process on Al alloys have been conducted. I cannot evaluate the novelty of this study.

(2) The quality of images is not high, especially for the images about microstructure observation.

(3) About the EBSD results, only the grain morphology were shown in Figs. 8 and 9. Hence, why did the authors perform EBSD analysis? If you only show the grain morphology, an OM is enough. It is expected to see more EBSD results and related deep discussion.

(4) The section of conclusions is too long.

Author Response

Response to Reviewer 2 Comments

Comments and Suggestions for Authors

This paper investigated the SCC behavior of 5083 Al alloy processed by Equal Channel Angular Pressing (ECAP). Although some results were reported and discussed, I cannot recommend the acceptance of this paper due to the following reasons.

(1) 5083 Al is a common Al alloy, and ECAP is not a novel processing method. I believe that plenty of studies about the ECAP process on Al alloys have been conducted. I cannot evaluate the novelty of this study.

Ans. In most of the journal the corrosion behaviour and the mechanical properties were studied after ECAP treatment. But the effect of SCC was not studied after ECAP so extensilely specially for this alloy. So this study is quite different from others.

(2) The quality of images is not high, especially for the images about microstructure observation.

Ans. The microstructure given here are quite clear and with high magnification.

(3) About the EBSD results, only the grain morphology were shown in Figs. 8 and 9. Hence, why did the authors perform EBSD analysis? If you only show the grain morphology, an OM is enough. It is expected to see more EBSD results and related deep discussion.

Ans. The structure is more prominent in EBSD results. In this paper many results are given. So other data obtained from the EBSD will be given in another paper.

(4) The section of conclusions is too long.

Ans. Many experiments were done in this particular paper. That’s why the conclusion section is little bit long. But little bit trimmimg are done.

Reviewer 3 Report

This paper studied the SCC behavior of ECAPed Al alloys in air and a 3.5% NaCl solution. Some methods and results require further validation and the suggestions are given as follows.

  1. Lines 203-205: The grain size are reported. Authors have to describe how to determine the grain size and analyze the uncertainty and accuracy of the reported grain size.
  2. In Figures 3 and 4, the unit of stress should be "MPa". 
  3. In Table 2, authors have to explain how to determine the "% error" and how many tests were conducted with each sample and what is the standard deviation for each "% error".
  4. A 3.5 wt% NaCl solution was employed as one of the test conditions. What is the reason to select this concentration of NaCl solution? 
  5. In Figure 5, Mg2Si precipitates were pointed out. Authors have to mention how to detect this phase and which instrument was used.
  6. Line 411: Authors mentioned grains 25, 26, 29, and 30. However, no indications were made in Figure 9.
  7. Lines 412-415: Authors reported the percentages (95% and 36%) of grains and the misorientation angles (10, 50, and 60 degrees). This information is quite obscure. Authors have to describe how to determine these data and the uncertainty and accuracy. 

Author Response

Response to Reviewer 3 Comments

Comments and Suggestions for Authors

This paper studied the SCC behavior of ECAPed Al alloys in air and a 3.5% NaCl solution. Some methods and results require further validation and the suggestions are given as follows.

Lines 203-205: The grain size are reported. Authors have to describe how to determine the grain size and analyze the uncertainty and accuracy of the reported grain size.

Ans. This determination was done using image J software.

In Figures 3 and 4, the unit of stress should be "MPa".

Ans. Necessary changes are done.

In Table 2, authors have to explain how to determine the "% error" and how many tests were conducted with each sample and what is the standard deviation for each "% error".

Ans. Each tests were conducted for three times and using software and standard deviation method the error was calculated. Correlation coefficient (R) is a statistical tool that provides information on the strength of linear relationship between the experimental and predicted values. The average absolute error is a quantity used to measure how close the prediction values are to the experimental ones.

Reference: Mechanical properties of Austenitic Stainless Steel 304L and 316L at elevated temperatures Raghuram Karthik Desua, Hansoge Nitin Krishnamurthy, Aditya Balua, Amit Kumar Guptaa, Swadesh Kumar Singh, Journal of Materials Research and Technology,

A 3.5 wt% NaCl solution was employed as one of the test conditions. What is the reason to select this concentration of NaCl solution?

Ans. This is the most common solution to conduct this kind of test finding from literature review.

In Figure 5, Mg2Si precipitates were pointed out. Authors have to mention how to detect this phase and which instrument was used.

Ans. Here the XRD was done. The XRD graph is given below.

                      As received                                                       ECAP 1

                      ECAP 2                                                               ECAP 3

Line 411: Authors mentioned grains 25, 26, 29, and 30. However, no indications were made in Figure 9.

Ans. Necessary changes are done.

Lines 412-415: Authors reported the percentages (95% and 36%) of grains and the misorientation angles (10, 50, and 60 degrees). This information is quite obscure. Authors have to describe how to determine these data and the uncertainty and accuracy.

Ans. This data are obtained from EBSD 

Round 2

Reviewer 2 Report

The authors should focus on the improvement of the paper.

Unfortunately, no useful reply was provided.

Author Response

Necessary changes have done and colored. We have also briefly added as per your suggestions.

Reviewer 3 Report

The revised manuscript can be accepted for publication.

Author Response

(The authors gave the same response as above.)
